# Oil Extracted *Moringa peregrina* Seed Cake as a Feed Ingredient in Poultry: A Chemical Composition and Nutritional Value Study

**DOI:** 10.3390/ani12243502

**Published:** 2022-12-12

**Authors:** Mohammed A. Al-Harthi, Youssef A. Attia, Mohamed F. Elgandy, Fulvia Bovera

**Affiliations:** 1Agriculture Department, Faculty of Environmental Sciences, King Abdulaziz University, P.O. Box 80208, Jeddah 21589, Saudi Arabia; 2Department of Veterinary Medicine and Animal Production, University of Napoli Federico II, via F. Delpino, 1, 80137 Napoli, Italy

**Keywords:** *Moringa peregrina* seed meal, chemical profile, protein quality, amino acids, metabolizable energy, chicken

## Abstract

**Simple Summary:**

This research evaluates the nutritional values, protein quality, amino acid utilization, and metabolizable energy of oil-extracted *Moringa peregrina* seed meal as a new protein resource for poultry nutrition. The results reveal that oil-extracted *Moringa peregrina* seed meal is a good source of nutrients and antioxidants and has 131.4 mg/100 g dry-weight tannic acids. The oil-extracted *Moringa peregrina* seed meal has a considerable amount of neutral detergent fiber, acid detergent fiber, and hemicellulose. Most of the total fatty acids were unsaturated, amounting to 84.57%. In conclusion, oil-extracted *Moringa peregrina* seed meal has a considerable level of bioactive materials, amino acids, fatty acids, minerals, and metabolizable energy. Nonetheless, its protein quality was lower than soybean meal protein, which suggested further research.

**Abstract:**

The chemical composition, antioxidant activity, tannic acid content, mineral, fatty acid, and amino acid profiles of oil-extracted *Moringa peregrina* seed meal (OEMPSM) were determined. Apparent (AME) and true (AMEn) metabolizable energy and apparent (AAAU) and true (TAAU) amino acid utilization were evaluated using a precision feeding trial. The protein (CP) quality was evaluated by a total efficiency analysis method. The antioxidant activity, gauged by 2,2-diphenyl-1-picrylhydrazyl (DPPH), was 237, 353, and 15.2 mg/mL for the water and ethanol extracts, and ascorbic acid, respectively. Tannic acids were 131.4 mg/100 g dry weight. The OEMPSM had 27.2% CP and 22.4, 15.1, and 15.8 MJ/kg of gross energy, AME and AMEn, respectively. The neutral detergent fiber, acid detergent fiber, and hemicellulose were 40.2, 29.7, and 10.5% DM, respectively. The 15.41% of total fatty acids were saturated and 84.57% unsaturated. The AAAU and TAAU of OEMPSM were 30.92% and 61.06%, respectively. From findings, OEMPSM comprises a valuable level of bioactive substances, amino acids, fatty acids, minerals, and energy; it can provide up to 1.12% of the requirements of total amino acids of chickens (1–21 days); however, the quality of its protein was found to be 44.6% less than that of protein of soybean meal.

## 1. Introduction

According to the estimation of the FAO [1], in 2017, 22 billion of chickens were consumed in the World, and this amount is expected to increase proportionally with the increase in the human population [2]. Soybean meal and maize are the two main ingredients in poultry diets and their cost largely increased in the last years due to the increased global demand; the recent limitations due to the COVID-19 pandemic [3] have brought out even more the risks of dependence on imported feeds. In addition, the feed prices spiked up due to the recent Russia–Ukraine war. A possible solution could be the use of local feed resources in poultry diet, using ingredients able to improve growth, health and meat quality [4], limiting or avoiding the use of antibiotics, harmful for human consumption.

*Moringa peregrina* (MP) is a plant from the family of Moringaceae, which comprises 13 species. In the Peninsula of Arabia, MP is the most common species of Moringa [5], but it is very poorly investigated for its chemical characteristics and nutraceutical properties compared with Moringa oleifera (MO) [6]. In the Kingdom of Saudi Arabia, MO Fiori is naturally present in the Dead Sea area, lower Jordan valley, Wadis Araba and Feynan. Locally known as Al-Ban or Al-Yassar, it is a 5–15 m tree with long seed pods, each holding 15–20 seeds. The seeds provide pivotal nutritional, economic, and medicinal values; the Bedouins use the extracted seed oil in cooking. Moreover, the intact seeds act as laxatives in medicine and livestock feed [7]. The MP seeds’ protein function characteristics, fatty acid composition, chemical composition, physicochemical characteristics, and medicinal uses were rarely evaluated in previous research. Besides its rich oil content (42–54%), MP seeds have a high level of protein and some vitamins and minerals. Its meal contains higher fat than soybean, though it has lower proteins, carbohydrates, and ash [8,9].

Al-Dabbas et al. [10] and Al-Dabbas [11] investigated the chemical-nutritional characteristics besides the antioxidant activity of different extracts from MP plants. Recently, there has been a rising interest in MP and some published studies investigating its pharmacological properties [12], the ability to increase immune defense [13], and the ability to act against multidrug-resistant bacterial strains and clinical fungus [14].

Therefore, this research aims to assess the chemical-nutritional profile of oil extracted *Moringa peregrina* seeds meal (OEMPSM) by deepening some aspects of the utilization efficiency of energy and amino acids to evaluate whether the product can be used as an ingredient in poultry feeding and act also as a functional feed.

## 2. Materials and Methods

The study was conducted at the Hada Al-Sham Research Station, Faculty of Environmental Sciences, King Abdulaziz University, Jeddah, Saudi Arabia. The research includes two parts: the first part, where the oil extracted MP seeds were analyzed for chemical characteristics and antioxidant property; and the second part, where the efficiency of utilization of energy and amino acids from oil extracted MP seeds were evaluated in an in vivo trial, involving broiler chickens. For the latter trial, the methodology was verified by the Committee of the Department of Arid Land Agriculture, King Abdulaziz University which regulates the rights and welfare of animals according to government law No. 9 dated 24 August 2010 and institutional approve code ACUC-22-1-2.

The meal from oil extracted MP seeds was purchased from Makkah in Saudi Arabia. The seeds were machine pressed for a cold press of oil extraction at ~40 °C, and the by-products were sun-dried and ground into powder (particle size 0.5 mm); then, they were kept in bags at room temperature until further analysis.

The feed extract’s antioxidant activity was analyzed using the DPPH (2,2-diphenyl-1-picrylhydrazyl) radical scavenging assay according to Shimada et al. [15] with some modifications. The water and ethanol extractions were performed at a temperature of 80 °C for 15 min. After the extraction, the samples were lyophilized. Around 1 mL of a solution with varying Moringa extract concentrations was mixed with 1 mL of 0.2 mM DPPH in methanol. The mixture was shaken and left in the dark at room temperature for half an hour. The solution’s absorbance at 517 nm was gauged using a spectrophotometer (model: SL273, OSAW, Ambala Cantt, India). For this assay, ascorbic acid was used as the standard. For the negative control (blank), the same amount (1 mL) of the extract was acquired from the control samples. All assays took place in triplicate. DPPH radical’s inhibition was evaluated as follows:DPPH radical inhibition % = (A blank − A test)/(A blank) × 100
where A blank is the absorbance of the control solution and A test is the absorbance of the test extract. The IC50 value (µg extract/mL) is the inhibitory concentration of the test content at which DPPH radicals were 50% scavenged and was calculated by interpolation from linear regression analysis.

As an index of polyphenols, total tannins were determined according to Burns [16] using an HPLC assay (Thermo3000, Waltham, MA, USA, wavelength 280 nm, at 28 °C).

Dry matter, crude protein, ether extract, crude fiber, and ash were determined, in triplicate, according to the following methods [17], respectively: 934.01, 954.01, 920.39, 954.18, and 942.05. Neutral and acid detergent fiber and hemicellulose were gauged according to Van Soest et al. [18]. Nitrogen-Free Extracts (NFE) were calculated as follow:dry matter − (protein + ether extract + crude fibre + ash)

Organic matter was calculated as 100-ash. Macro- and microminerals were assessed via atomic absorption (Avanta Z, GBC Scientific Equipment Ltd., Braeside, Australia) by utilizing a standard curve according to Jackson [19]; phosphorus was measured according to Dickman and Bray [20].

Total lipids of OEMPSM were extracted according to Folch et al. [21] and their fatty acid profile was measured according to Radwan [22], using a Shimadzu GC4CM gas chromatograph (Shimadzu Corp., Kyoto, Japan), with field influence mobility (pFE), a glass column (3 m × 3.1 mm ID, cat. no. 221-14368-31, Shimadzu Corp., Kyoto, Japan) packed with 5% Diethylene Glycol Succinate (DEGS) and equipped with a Flame Ionization Detector (FID). Two determinations were conducted for every test, and the average was measured.

Apparent metabolizable energy and its values corrected for nitrogen (AME and AMEn) and apparent and true amino acid utilization (AAAU, TAAU) of OEMPSM were measured as suggested by Sibbald [23] and described by Borin et al. [24]. A total of 14, 16-week-old adult Single-Comb White Leghorn roosters, equally divided into 2 groups, were housed in individual cages (35 × 30 × 40 cm). Each cage was provided with one stainless steel nipple drinker and exposed to 16:8 light–dark cycle. The two groups were left to fast for a day to eliminate the possibility of feed residues present in their gut. One group was then forced-fed 30 g of the OEMPSM once per rooster and the excreta were collected over the following day. The other group was left to fast for a second day to evaluate the endogenous losses. Excreta were collected separately from both groups using the total feces from each rooster. Feathers were removed from the excreta, and they were oven-dried at 70 °C until they reached a constant weight. Then, the excreta were equilibrated with humidity for a day, weighed, ground into a dry powder, and sieved to a particle size of 0.5 mm. The dried samples were kept in a tightly sealed glass bottle until analysis. Duplicate samples were allocated for the determination of energy concentrations and amino acid profiles in the excreta, OEMPSM, and endogenous losses. Then, the average was gauged. Energy was gauged via an adiabatic bomb calorimeter (model C400, IKA Analysentechnik GmbH, Gribheimer Weg 5, D79423, Heitersheim, Germany).

The amino acid content of OEMPSM was measured at the Novus International Research Center, St Charles, MO, USA, per AOAC [17] using a Hitachi L8900 amino acid analyzer (Minato-ku, Tokyo, Japan). Amino acids’ percentage in the crude protein content of OEMPSM and excreta of fed and fasted-control were also calculated.

The apparent amino acid utilization was calculated as a ratio between amino acid intake and amino acids excreted by fed-group (amino acid in OEMPSM × amount feed intake) − (amino excreted by fed group (amino acid in excreta × amount of excreta voided)/amino acid intake (amino acid in OEMPSM × amount feed intake).

The true amino acid utilization was calculated as a ratio between amino acid intake and amino excreted correct for endogenous amino acids loss (amino acid in OEMPSM × amount feed intake) − (amino excreted by fed group (amino acid in excreta × amount of excreta voided) − amino excreted by fasted-group (amino acid in excreta of fasted-control × amount of excreta voided of fasted-control)/amino acid intake (amino acid in OEMPSM × amount feed intake). The calculation was performed on dry matter basis of feed and excreta. Moreover, the ratio of total amino acids of OEMPSM to total amino acids needed by broiler chickens from day 1 to day 21 of life [25] was measured.

Protein quality was evaluated according to Woodham [26]. A growth trial was conducted on 60 male Ross 308 broiler chickens from 14 to 28 days of age (14 days of trial). The broilers were divided into two groups (6 replicates per group, 5 male broilers per replicate) and housed in metal cages of 45 × 55 × 60 cm (5 broilers, 1 replicate per cage) provided with one-tube feeder and one-nipple waterers. Water and mash experimental feed were provided ad libitum. Broiler chickens were raised per standard veterinary prescriptions and husbandry conditions. The groups were submitted to different dietary treatments. The first group was fed a maize-soybean meal (control), the second one was fed an isonitrogenous and isoenergetic diet containing OEMPSM. In detail, the OEMPSM provided 42.3% of the total protein offered by the main protein source (soybean meal) in the control diet. The chemical-nutritional characteristics of the diets [17] are reported in Table 1. The diets were formulated according to NRC [25]. Within 1–13 days of age, chickens were fed a commercial starter diet (20% CP and 12.52 MJ of Metabolizable Energy (ME), 1% Ca, and 0.50% available phosphorus) on a 23:1 light–dark cycle. Chicks were weighed based on replicates at 14 and 28 days of age, and feed intake was estimated. Bodyweight gain and feed and protein intakes were calculated per chicken per day. Protein efficiency was then gauged by dividing protein intake by body weight gain.

### Statistical Analysis

The normality of the data and error distribution was evaluated using the with Shapiro–Wilk test SAS Institute Inc. SAS/STAT Software, version 9; SAS Institute, Inc.: Cary, NC, USA [27]. The random selection of the animals and samples assured the respect of the 4 assumptions of analysis of variance (ANOVA). The homoscedasticity (variance homogeneity) was evaluated by Levene’s test SAS^®^ [27].

Data on chemical characteristics and on bioassay of energy metabolizable and amino acid utilization of the *Moringa* were not statistically evaluated. Data on bioassay of protein quality using in vivo trial (broiler weight gain, feed intake, protein intake, and efficiency of total protein) were analyzed with the SAS^®^ [27] software package using a one-way ANOVA (GLM procedure) according to the following model: Yij = m + Di + eij, where Y is the single observation, m the general mean, D the effect of the diet (I = control or OEMPSM), e the error. Mean comparison took place using the Tukey’s test [27] considering significant *p*-values lower than 0.05.

## 3. Results

Table 2 shows the DPPH scavenging activity measured for water and ethanol extracts and ascorbic acid, as well as the tannic acid content of OEMPSM. The 2,2-diphenyl-1- picrylhydrazil radical was high in the ethanol extract (353.2 ± 0.28 mg/mL). Table 3 indicates the chemical profile of OEMPSM compared with *Moringa oleifera*, maize grain, and soybean meal. In general, OEMPSM had lower ash content (3.93%) and high ADF content (29.73%) than MO. The CP content of OEMPSM was about 49.2% of that of soybean meal. The mineral composition shown in Table 4 highlights the lower P (2.1 g/kg), K (0.052 g/kg), and Cu (0.010 g/kg) content of OEMPSM compared with MO, unlike the rich source of Ca.

The fatty acid profile of OEMPSM (Table 5) falls within the range available in the literature for MP, which confirms the higher C18:3 compared with MO. OEMPSC has higher proportion of palmitic acid (16:0) and lower stearic acid (18:0) than MO as these are the second and third more abundant fatty acids after oleic acid, respectively. It was not possible to compare the percentage of C20:4 n-6 fatty acid due to lack of data in the literature.

The amino acid profile of OEMPSM (Table 6) shows low levels of methionine and lysine compared to broiler requirements; however, the level of methionine + cysteine meets the requirements of broilers aged from 0 to 3 weeks.

Table 7 indicates the apparent and true digestibility of amino acids (AA) of OEMPSM. Arginine showed the lowest true digestibility (28.1%), while the highest was found for isoleucine (80.1%). The OEMPSM showed high contents of AME and AMEn compared with MO defatted seeds and soybean meal (Table 8).

Table 9 reveals the growth performance of OEMPSM–fed broilers. Feed and protein intake were not affected by the feeding treatments; however, body weight gain was 33.4% (*p* < 0.01), which is smaller than that of the control group, while *Protein efficiency ratio* was 74% higher (*p* < 0.01).

## 4. Discussion

The OEMPSM had a moderately high content of tannins (131.4 mg/100 g DM), which may adversely affect chickens’ growth performance [55], as it inhibits proteins, amino acids, and minerals availability [56]. However, a recent review [57] indicated that a level of tannins between 0.5 and 5 g/kg can enhance the gut health and growth rate of poultry due to its antioxidant, anti-inflammatory, and antimicrobial properties. The proximate analyses in terms of organic matter, dry matter, CP, ether extract, crude fiber, NDF, ADF, and hemicellulose are within the range of levels cited in the available literature, indicating that OEMPSM is a rich source of nutrients for animals and humans [10,28,58].

The nutrient profile of OEMPSM is like those of other Moringa species [9,34,59]. The OEMPSM had, in general, a better nutrient profile than maize, but worse than that of soybean meal [25,32,34,35].

The mineral profile of OEMPSM is consistent with that of MO, as reported by El-Naggar et al. [31] and Ashour et al. [60]. The outcomes imply that OEMPSM is a rich source of Ca and Mn, Cu, Fe, and Zn, which are essential for bone and eggshell calcification [61].

It was found that OEMPSM is a rich source of essential fatty acids, especially C18:3, which is a beneficial fatty acid of important health impact, and it has a favorable ratio of UFA to SFA (5.48:1). The n-3 fatty acids is comparable to those of MP [10,51] and maize [25], while it was higher than that of MO [38,49,52] and lower than that of soybean meal [25,39]. The results also showed that OEMPSM is a rich source of MUFA (C18:1) and a beneficial source of PUFA (n-3 and 20:4 n-6). It is well known that the degree of saturation of dietary fat has an essential role in fat digestion, as UFA are more easily digested than SFA [40,61]. Therefore, a higher saturation level or lower UFA to SFA ratio can reduce the digestibility of lipid sources in broiler chickens [41]. Recently, Jimenez-Moya et al. [42] evaluated diets with different UFA to SFA ratios for broiler chickens (from 4.54 to 1.14). They found that fat saturation level affected lipid digestion more than other factors such as the free fatty acid content. The differences in single fatty acid levels among data reported in the literature could be due to method of oil extractions, product type (seeds, or pods vs. leaves), environmental, and agronomic conditions [10,28,61].

The OEMPSM’s protein was characterized by an adequate amount of essential amino acids, such as leucine, arginine, lysine, and valine. Moreover, valine, threonine, and isoleucine levels were like those reported by Heuzé et al. [43] for MO. Our results for methionine + cysteine (1.17), threonine (0.73), lysine (0.87), and glycine (1.60) were comparable to those reported for MO [32,38].

The amino acid profile, total amino acids, and ratios of total amino acids to total required amino acids in OEMPSM were higher than those of maize, and lower than in the soybean meal [25]. These differences could be attributed to the levels of crude protein (26.01 vs. 8.5 and 47.5, respectively, for OEMPSM, maize and soybean meal), the plant species, environmental and agronomic conditions, preparation methods, and product type (seed, meal, or leaf) [44,56].

The present findings indicate that OEMPSM can meet some of the AA needs of broiler chickens aged 1–21 days, except for threonine, methionine, isoleucine, lysine, histidine, and tryptophan [25]. Methionine was found to be the first (76%) and lysine to be the second (79%) limiting amino acid for broiler chickens at 1–21 days of age. The number of total amino acids in OEMPSM was more than that required for broilers (52.96% vs. 49.16%).

To date, there are no studies on amino acid utilization of OEMPSM. The AAAU of OEMPSM ranged from 8.72% for methionine to 62.04% for lysine, with an average of 30.92%. The corresponding values for TAAU were 46.02% for methionine and 71.54% for lysine, with an average amino acid utilization of 61.06%.

These outcomes imply that correction for endogenous amino acid losses (30.92% versus 61.06%) increased TAAU up to 97.5%. The lowest recovery of TAAU was observed for arginine (28.1%), and the highest was observed for isoleucine (80.1%), followed by threonine (77.7%), and valine (73.2%). A similar value (21.1% to 62.8%) of amino acid utilization was found by other authors [2,24]. The low apparent and actual utilization of arginine and serine from OEMPSM found in this study is not evident, but it could be attributed to mixing with other nutrient-hostile components of OEMPSM such as tannins and fiber fractions. The downside impacts of tannins on diets’ nutritional value for monogastric and ruminant animals have been mentioned before [45,46].

Estimated levels of AME and AMEn correspond to 67.7% and 70.6% of the total OEMPSM gross energy (22.37 MJ/kg). Results in the literature indicate that OEMPSM has comparable values to MO and maize grains, but it showed higher values than soybean meal [25,28,37,47]. To our knowledge, accurate apparent and true metabolizable energy values for OEMPSM were not previously discussed in the literature. However, it was evident that the metabolizable energy of OEMPSM exceeded (15.15 MJ/kg) that of soybean (10.96 MJ/kg). This high metabolizability of OEMPSM energy can be explained by its high organic matter (96.1) and ether extract content (9.94%) [25,47,48].

Furthermore, the results of amino acids utilization were confirmed by the OEMPSM protein’s quality assay. The results showed that protein efficiency ratio was significantly lower (74%) than that of soybean meal Table 9. These outcomes verify the findings of amino acid utilization and could be explained by the high dietary fiber content (32.23%) and especially by the tannin content (131.4 mg/g) in OEMPSM, as mentioned earlier [2,24]. Furthermore, tannins and dietary fiber adversely influenced feeds’ energy and nutrient utilization values [25,56]. The present results showed that OEMPSM had valuable nutrient profiles and could be a suitable feed resources for animals [38,41,43,45,47,52].

## 5. Conclusions

Based on our findings, the OEMPSM showed valuable level of nutrients (protein, essential amino acids, and minerals) adequate proportions of fatty acids and has a higher level of metabolizable energy. These characteristics are very interesting for monogastric animals, and thus OEMPSM could be considered as a potential ingredient for poultry diets. However, other in vivo and in vitro studies should be conducted to deeply evaluate the biological feed value of the OEMPSM.

## Figures and Tables

**Table 1 animals-12-03502-t001:** Ingredients (g/kg) and chemical-nutritional characteristics of the diets used in the Protein efficiency ratio assay.

Ingredients	Maize-Soybean Diet	OEMPSC Diet
Maize	714	655
Soybean meal (44%)	153	105
Wheat bran	98	105
OEMPSC	--	100
Dicalcium phosphate	20	20
Limestone	8.0	8.0
Sodium chloride	4.0	4.0
Premix ^a^	3.0	3.0
Chemical-nutritional characteristics		
ME MJ/kg diet, ^b^	11.96	12.10
CP, g/kg ^c^	135.8	138.7
Ca, g/kg ^b^	8.79	10.4
Non-phytate phosphorus, g/kg ^b^	4.38	4.47
Methionine, g/kg ^b^	3.22	3.22
Sulfur amino acids, g/kg ^b^	5.94	6.32
Lysine, g/kg ^b^	6.36	5.84
Ether extract, g/kg ^c^	28.3	35.3
Crude fibre, g/kg ^c^	35.1	63.6
Ash, g/kg ^c^	51.3	53.5

^a^ Vit + Min mix. provides per kilogram of the diet: Vit. A, 12,000 IU, vit. E (dl-α-tocopheryl acetate) 20 mg, menadione 2.3 mg, Vit. D3, 2200 ICU, riboflavin 5.5 mg, calcium pantothenate 12 mg, nicotinic acid 50 mg, Choline 250 mg, vit. B_12_ 10 g, vit. B_6_ 3 mg, thiamine 3 mg, folic acid 1 mg, d-biotin 0.05 mg. Trace mineral (mg/kg of diet): Mn 80, Zn 60, Fe 35, Cu 8 and Selenium 0.1 mg. ^b^ Calculated composition, ^c^ Chemical composition. OEMPSC: oil extracted *Moringa peregrina* cake seeds.

**Table 2 animals-12-03502-t002:** Antioxidant activity extract and ascorbic acid (synthetic antioxidant) of oil extracted *Moringa peregrina* seed cake (mean ± standard deviation).

	DPPH Scavenging Activity (Inhibition Concentration; IC50 mg/mL Extract)
Water extract	237.26 ± 0.70
Ethanol extract	353.16 ± 0.28
Ascorbic acid	15.16 ± 0.30
Tannic acid, mg/100 g dry matter	131.4 ± 2.37

DPPH: 2,2-diphenyl-1-picrylhydrazyl.

**Table 3 animals-12-03502-t003:** Proximate chemical composition (%), on dry matter basis of oil extracted *Moringa peregrina* seed cake used in this study compared with *Moringa oleifera* cake, maize grain, and soybean meal (CP 48.5%) in the literature.

	OEMPSC	*Moringa oleifera*	Maize Grain	Soybean Meal
Dry matter, DM	95.79	89.21–94.97 [28,29]	87.20	88.00
Ash	3.93	4.30–10.53 [29,30]	1.50	7.30
Crude protein, CP	27.15	25.00–53.49 [29,31]	9.70	55.20
Ether extract, EE	9.94	3.22–18.28 [29,30]	4.20	1.70
Crude fibre, CF	32.23	13.65–30.81 [29,32]	2.60	4.40
NDF	40.19	8.20–44.40 [31,33]	13.20	10.50
ADF	29.73	6.70–27.10 [31,33]	4.40	5.70
Hemicellulose *	10.46	5.69–17.30 [31,34]	8.80 *	4.80 *
NFE *	30.96	10.72–57.77 [34,35]	82.00	31.40
Organic matter **	96.07	87.13–94.80 [31,35]	85.70 *	80.70 *

OEMPSC: oil extracted *Moringa peregrina* seed cake; * calculated as NDF-ADF; NFE: nitrogen free extracts, calculated as DM − CP − EE − CF + Ash; ** calculated as: OM = 100 − Ash.

**Table 4 animals-12-03502-t004:** Mineral’s composition (g/kg), as-fed basis, of oil extracted *Moringa peregrina* seed cake used in this study compared with *Moringa oleifera* cake, maize grain, and soybean meal (CP 48.5%) in the literature.

	OEMPSC	*Moringa oleifera*	Maize Grain	Soybean Meal
Calcium	17.500	0.250–130.120 [29,36]	0.610	3.432
Phosphorus	2.100	7.053–26.000 [36,37]	2.965	6.248
Sodium	9.800	0.184–14.753 [29,33,37,38,39,40,41,42,43,44,45,46,47,48]	0.087	0.114
Potassium	0.052	0.570–14.510 [29,36]	3.750	21.384
Magnesium	4.300	0.258–210.240 [29,36]	1.221	2.816
Manganese	1.024	0.004–1.370 [29,36]	0.011	0.039
Copper	0.010	0.079–2.070 [36,37]	0.004	0.015
Iron	0.290	0.037–8.350 [29,36]	0.689	0.177
Zinc	0.026	0.012–8.800 [29,36]	0.030	0.050

OEMPSC: oil extracted *Moringa peregrina* seed cake.

**Table 5 animals-12-03502-t005:** Fatty acids profile (% of total fatty acids) of oil extracted *Moringa peregrina* seed cake used in this study compared with *Moringa peregrina* and oleifera seeds, maize grain, and soybean meal (CP 48.5%) oils in the literature.

	OEMPSC	*Peregrina*	*Oleifera*	Maize Grain	Soybean Meal
C6:0	0.102	-		0.110	0.050
C13:0	0.230		-	-	-
C14:0	0.243	-	0.080–0.150 [49,50]	0.080	0.140
C14:1	0.254	0.080–0.620 [10,51]	-	-	-
C15:0	0.125	-	-	-	-
C15:1	0.160	-	-	-	-
C16:0	9.290	-	0.500–6.440 [38,52]	11.250	13.060
C16:1	1.804	8.540–15.340 [9,51]	0.500–1.920 [38,52]	0.140	0.130
C17:0	0.109	1.540–4.950 [50,51]	0.080–0.090 [38,49]	0.040	0.130
C18:0	3.497	0.120–0.160 [10,49]	4.450–7.940 [38,49]	2.070	4.080
C18:1	77.506	3.080–9.750 [10,51]	64.560–79.580 [9,38]	23.740	16.190
C18:2	0.632	57.530–78.000 [51,53]	0.580–5.790 [34,38]	59.410	55.200
C18:3	1.742	0.420–15.320 [49,50]	0.0–0.170 [38,49]	1.620	10.090
C20:0	1.835	0.050–3.420 [10,51]	1.570–5.100 [38,39]	0.290	0.290
C20:4	2.469	1.730–4.940 [10,51]	-	-	-
TFA	99.998	-	-	-	-
SFA	15.431	-	-	-	-
UFA	84.567	-	-	-	-
MUFA	79.724	-	-	-	-
PUFA	4.843	-	-	-	-
UFA/SFA	5.48	-	-	-	-

OEMPSC: oil extracted *Moringa peregrina* seed cake. TFA: total fatty acids; SFA: saturated fatty acids; UFA: unsaturated fatty acids; MUFA: mono-unsaturated fatty acids; PUFA: poly-unsaturated fatty acids.

**Table 6 animals-12-03502-t006:** Protein content and amino acids profile (%), as-fed basis, of oil extracted *Moringa peregrina* seed cake used in this study compared with *Moringa oleifera* cake, maize grain, and soybean meal (CP 47.5%) in the literature.

	OEMPSC	MODS	Maize Grain	Soybean Meal	Broiler Requirements
Crude protein	26.01	26.8–40.0 [30,32]	8.50	47.50	23.00
Threonine	0.73	0.60–1.61 [30,32]	0.29	1.87	0.80
Valine	0.96	1.10–2.31 [30,32]	0.40	2.22	0.90
Methionine	0.38	0.50–0.61 [30,32]	0.18	0.67	0.50
Cystine	0.79	0.60 [30]	0.18	0.72	0.90
Met + Cys	1.17	1.10–1.21 [30,32]	0.36	1.39	0.80
Isoleucine	0.76	0.90–1.83 [30,32]	0.29	2.12	1.20
Leucine	1.10	1.60–5.25 [30,32]	1.00	3.74	0.72
Phenylalanine	0.77	1.54–4.84 [32,38]	0.38	2.34	1.34
Tyrosine	0.65	0.77–2.45 [32,38]	0.30	1.95	0.35
Phen + Tyr	1.42	2.31–7.29 [32,38]	0.68	4.29	1.10
Histidine	0.44	0.80–2.07 [32,38]	0.23	1.28	1.25
Lysine	0.87	0.57–3.97 [32,38]	0.26	2.96	1.25
Arginine	1.44	2.55–5.22 [32,38]	0.38	3.48	1.11
Glycine	1.60	0.97–1.83 [32,38]	0.33	2.05	NR
Serine	0.75	1.04–2.41 [32,38]	0.37	2.48	NR
Glycine + Serine	2.35	2.01–4.24 [32,38]	0.70	4.53	1.25
Tryptophan	0.39	1.1 [54]	0.86	1.38	0.90

OEMPSC: oil extracted *Moringa peregrina* seed cake; MODS: *Moringa oleifera* defatted seeds; Met + Cys: methionine + cystine; Phen + Tyr: phenylalanine + tyrosine; NR: not reported.

**Table 7 animals-12-03502-t007:** Apparent and true amino acids utilization (%) of oil extracted *Moringa peregrina* seed cake used in this study.

	Apparent	True
Threonine	40.84	77.74
Valine	45.60	73.20
Methionine	8.72	46.02
Cystine	26.20	72.90
Isoleucine	38.56	80.06
Leucine	43.28	66.68
Phenylalanine	27.56	62.56
Tyrosine	31.40	49.60
Histidine	14.72	45.52
Lysine	62.04	71.54
Arginine	7.20	28.10
Glycine	42.24	71.74
Serine	13.60	48.10
Average	30.92	61.06

**Table 8 animals-12-03502-t008:** Gross energy (GE), apparent metabolizable energy (AME), and apparent metabolizable energy corrected to zero nitrogen retention (AMEn), as MJ/kg DM, of oil extracted *Moringa peregrina* seed cake used in this study compared with *Moringa oleifera* cake, maize grain, and soybean meal (CP 48%) in the literature.

MJ/kg DM	OEMPSC	*Moringa oleifera* Defatted Seed	Maize Grain	Soybean Meal
GE	22.45 ± 0.179	19.40 [37]	18.62 [28]	19.47 [28]
AME	15.20 ± 0.110	14.60 [37]	15.85 [28]	10.96 [28]
AMEn	14.65 ± 0.150	-	15.56 [28]	10.43 [28]

OEMPSC: oil extracted *Moringa peregrina* seed cake.

**Table 9 animals-12-03502-t009:** Protein efficiency ratio of oil extracted *Moringa peregrina* seed cake compared to the control-based maize and soybean meal oil.

	Control	OEMPSC	SEM	*p*-Value
Initial body weight, g	416	419	28.6	0.436
Final body weight,	1412	1084	47.5	0.001
Body weight gain, g	71.3 ^a^	47.5 ^b^	4.62	0.001
Feed intake, g	92.2	107.0	5.43	0.123
Protein intake, g	12.91	14.98	1.37	0.078
Protein efficiency ratio	0.181 ^b^	0.315 ^a^	0.087	0.0001

^a,b^ Means within a row not sharing a common superscript are significantly different (*p* < 0.05). SEM = Standard error of mean, *p* value = Probability level; OEMPSC: oil extracted *Moringa peregrina* seed cake.

## Data Availability

Data can be required from the corresponding authors.

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
