# Peer review of "Oil Extracted Moringa peregrina Seed Cake as a Feed Ingredient in Poultry: A Chemical Composition and Nutritional Value Study"

_animals, 2022, doi:10.3390/ani12243502_

Round 1

Reviewer 1 Report

This an interesting work assessing the potential use of oil extracted Molinga pelegrina seed cake as feed ingredient in poultry diets. The manuscript is written with a logic structure and the Introduction section summarizes well the theoretical background of the research. The methodology used is adequately described and appropriate. In my opinion, the work would be much more interesting if more complete information about productive performance, blood parameters, renal and hepatic performance of the broiler chicken's would be measured at the end of the feeding trial.

The experimental design of the feeding trial is somewhat confusing and the authors should clearly state the initial age of the broilers, their average weight and the duration of the experiment, as well as broiler's final weight.

Some parts of the Results section are repetitive and certain results and comments are not consistent with data displayed in the tables. This section should partially be re-written and several Tables better explained (please, see the .pdf revised version of the MS).

The authors should check the use of italics in scientific names along the whole MS, including the title and the reference list.

Some other comments and suggestions are detailed in the .pdf revised version of the MS.

Author Response

This an interesting work assessing the potential use of oil extracted Molinga pelegrina seed cake as feed ingredient in poultry diets. The manuscript is written with a logic structure and the Introduction section summarizes well the theoretical background of the research. The methodology used is adequately described and appropriate. In my opinion, the work would be much more interesting if more complete information about productive performance, blood parameters, renal and hepatic performance of the broiler chicken's would be measured at the end of the feeding trial.

Au: thank you very much for your perfect comment. Unfortunately, in this trial, however it is of added value, it was not possible to evaluate blood or renal and hepatic parameter. We hope in a further trial we’ll can do as this article focus on the chemical composition, antioxidant properties, ME, protein quality and apparent and true amino acids utilization.

.

The experimental design of the feeding trial is somewhat confusing, and the authors should clearly state the initial age of the broilers, their average weight, and the duration of the experiment, as well as broiler's final weight.

Au: revised and Indications and results have been added.

Some parts of the Results section are repetitive and certain results and comments are not consistent with data displayed in the tables. This section should partially be re-written and several Tables better explained (please, see the .pdf revised version of the MS).

Au: thank you very much for your comment. Needs further revision Revised

The authors should check the use of italics in scientific names along the whole MS, including the title and the reference list.

Au: thank you very much for your comment, revised

 Some other comments and suggestions are detailed in the .pdf revised version of the MS.

Au: thank you very much for your comment. Revised.

All the changes indicated in your pdf file have been applied. Thank you.

Reviewer 2 Report

Oil Extracted Moringa Peregrina Seed Cake as a Feed Ingredient in Poultry Diet: A Chemical-Nutritional Evaluation

Dear Authors,

manuscript describes possibility use of oil-extracted Moringa peregrina seed meal (OEMPSM) in nutrition of broiler chickens. The component itself is an interesting solution from the nutritional point of view. It could be better suited to feeding ruminants, but can be use also in poultry nutrition as a source of dietary fibre or active substances. In the presented diets, the high content of crude fibre (over 300 g * kg-1) and tannins limiting the action of digestive enzymes could cause such low gains of chickens during the experiment (Table 9). The research itself focused mainly on indicating the chemical composition and nutritional value as a result of replacing part of the total protein of soy with this by-product, but with too much CF and tannins, 100 g * kg-1 of the diet limits the average daily gains of chickens, maybe in the future it is worthwhile limit the level of OEMPSM to 50 g * kg-1 and take into account the greater share of CP in the diet (as e.g. for Ross 308 in starter diets equal to 200 g * kg-1). The article is interesting and gives a broad picture of the chemical composition and antioxidant properties of OEMPSM, however, in order to increase the cost-effectiveness of its application, it is necessary to perform additional in vivo and also in vitro tests mentioned in the conclusions, as a result of which it should be possible to more accurately determine the maximum share of OEMPSM, without so significant differences in the daily weight gain of chickens. Below are some comments and comments that are helpful in the process of proofreading the text:

Line 1

In title of manuscript is: A chemical-nutritional evaluation, maybe better to describe as: A chemical composition and nutritional value.

Line 19

Unit for detergent fibre fractions is needed. Perhaps % in DM.

Line 153

Small contradiction, because in Table 1 is used maize and corn (UK and American). Following Linnaeus’ point of view, perhaps maize will be better solution in this case (maize-soyabean diet)

and chemical-nutritional characteristics, in my opinion chemical composition will be simpler and better as a description, and of course it could be calculated indirectly basis on ingredients chemical composition or directly due to chemical analysis of whole mixture of diet.

Line 157

In text of manuscript is: “…bCalculated composition and cchemical composition…” When we assume that chemical composition of diet describes whole content of nutrients in it, then each nutrient is a part of the diet and in this case better is to use calculated values and values determined analytically for each nutrient.

Line 160-161

“…Data on chemical-nutritional characteristics and on bioassay of energy metabolizable and amino acid utilization of the moringa were not statistically evaluated…”.

Maybe better is to change chemical-nutritional on chemical and moringa on Moringa sp. (italicise needed).

Line 166

I wondering if Tukey’s post hoc test is not too conservative. There are two treatments maybe Duncan’s test could be enough.

Line 171 and 182

In text is: Moringa oleifera, italicise needed: Moringa oleifera (binomal nomenclature of species is needed).

Line 187-189

Italicise needed in the title of table and in 3rd column of Table 3 (heading).

Line 196-198

Italicise needed in title of table and in 3rd column of Table 4 (heading).

Line 203-205

Italicise needed in title of table. Modification and italicise also needed in 3rd and 4th column of Table 5 (heading) with whole binominal nomenclature or using only M. peregina and M. olifera.

Line 214-215

Italicise needed in title of Table 6. Adjustment of font and width of columns in table.

Line 222

In table 7 digestibility of amino acids is showing, in my opinion more information in Materials and Methods are needed to describe how it was determined. Because in this case will be difficult to obtain results for broiler chickens fed only OEMPSM by 4-5 days with quite high level of tannins (differential method or regression equation or it is possible?).

Line 225-228

Italicise needed in the title of table and in 3rd column of Table 8 (heading).

Line 236

P-value instead of P value

In manuscript is: Total protein efficiency, but often is used Protein efficiency ratio (PER), maybe it will be more appropriate.

Line 445

References

Lack of articles: from 55 to 61 in the Discussion.

Author Response

Dear Authors,

This Manuscript describes possibility use of oil-extracted Moringa peregrina seed meal (OEMPSM) in nutrition of broiler chickens. The component itself is an interesting solution from the nutritional point of view. It could be better suited to feeding ruminants, but can be use also in poultry nutrition as a source of dietary fibre or active substances. In the presented diets, the high content of crude fibre (over 300 g * kg-1) and tannins limiting the action of digestive enzymes could cause such low gains of chickens during the experiment (Table 9). The research itself focused mainly on indicating the chemical composition and nutritional value as a result of replacing part of the total protein of soy with this by-product, but with too much CF and tannins, 100 g * kg-1 of the diet limits the average daily gains of chickens, maybe in the future it is worthwhile limit the level of OEMPSM to 50 g * kg-1 and take into account the greater share of CP in the diet (as e.g. for Ross 308 in starter diets equal to 200 g * kg-1). The article is interesting and gives a broad picture of the chemical composition and antioxidant properties of OEMPSM, however, in order to increase the cost-effectiveness of its application, it is necessary to perform additional in vivo and also in vitro tests mentioned in the conclusions, as a result of which it should be possible to more accurately determine the maximum share of OEMPSM, without so significant differences in the daily weight gain of chickens. Below are some comments and comments that are helpful in the process of proofreading the text:

Au: thank you very much for your good comment. We hope in a further trial we’ll can do additional in vivo and also in vitro tests for deep understanding, however, the main objective of this work was to test chemical composition, antioxidant properties, ME, protein quality and apparent and true amino acids utilization.

  .

Line 1

In title of manuscript is: A chemical-nutritional evaluation, maybe better to describe as: A chemical composition and nutritional value.

Au: corrected

Line 19

Unit for detergent fibre fractions is needed. Perhaps % in DM.

Au: you are right, thank you

Line 153

Small contradiction, because in Table 1 is used maize and corn (UK and American). Following Linnaeus’ point of view, perhaps maize will be better solution in this case (maize-soyabean diet) and chemical-nutritional characteristics, in my opinion chemical composition will be simpler and better as a description, and of course it could be calculated indirectly basis on ingredients chemical composition or directly due to chemical analysis of whole mixture of diet.

Au: corn has been changed as maize all along the text

Line 157

In text of manuscript is: “…b Calculated composition and c chemical composition…” When we assume that chemical composition of diet describes whole content of nutrients in it, then each nutrient is a part of the diet and in this case better is to use calculated values and values determined analytically for each nutrient.

Au: thank you for your comment but we do not completely agree with your point. The measured value reflect the true chemical composition of the diets and it could be slightly different from the calculated value.

Line 160-161

“…Data on chemical-nutritional characteristics and on bioassay of energy metabolizable and amino acid utilization of the moringa were not statistically evaluated…”.

Maybe better is to change chemical-nutritional on chemical and moringa on Moringa sp. (italicise needed).

Au: done

Line 166

I wondering if Tukey’s post hoc test is not too conservative. There are two treatments maybe Duncan’s test could be enough.

Au: in our opinion Tukey’s post hoc test is appropriate because there are equal numbers of subjects contained in each group and addition in two treatments experiment, significant P value is adequate when exist

Line 171 and 182

In text is: Moringa oleifera, italicise needed: Moringa oleifera (binomal nomenclature of species is needed).

Au: done

Line 187-189

Italicise needed in the title of table and in 3rd column of Table 3 (heading).

Au: done

Line 196-198

Italicise needed in title of table and in 3rd column of Table 4 (heading).

Au: done

Line 203-205

Italicise needed in title of table. Modification and italicise also needed in 3rd and 4th column of Table 5 (heading) with whole binominal nomenclature or using only M. peregina and M. olifera.

Au: done

Line 214-215

Italicise needed in title of Table 6. Adjustment of font and width of columns in table.

Au: done

Line 222

In table 7 digestibility of amino acids is showing, in my opinion more information in Materials and Methods are needed to describe how it was determined. Because in this case will be difficult to obtain results for broiler chickens fed only OEMPSM by 4-5 days with quite high level of tannins (differential method or regression equation or it is possible?). The experiment was run with adult Leghorn 16-weeks roasters according to the standard procedure using the force-feeding assay.  The method of calculation of AAAU and TAAU were added. Regression will be considered in future experiment using different length of the experiment period and incremental amount of tested feed stuffs. Thank you for this perfect suggestion. 

Line 225-228

Italicise needed in the title of table and in 3rd column of Table 8 (heading).

Au: done

Line 236

P-value instead of P value

Au: done

In manuscript is: Total protein efficiency, but often is used Protein efficiency ratio (PER), maybe it will be more appropriate.

Au: corrected along the text

Line 445

References

Lack of articles: from 55 to 61 in the Discussion.

Au: the reported references are cited in the footnotes of the tables

Reviewer 3 Report

Dear authors, this study provides new and important data about the chemical characteristics and nutraceutical properties of Moringa peregrina (MP). Moreover, the use of the English language in the present study was appropriate and only some phrases must be rewritten. However, authors must answer several questions according mainly to the zootechnical and secondly to the statistical methods used. Firstly, regarding the in vivo experiment which was executed in the present study only n=60 birds were used, and this number of broilers is not appropriate to exert accurate results. Secondly, in the statistics section authors must answer if they check their data for normality with no parametric tests. Conclusively, the in vivo experiment must be excluded, and this study needs to be major revised to be published in this journal. The questions which must be answered and the changes which are proposed to be done are presented line by line in the following paragraphs.

Abstract:

---------------------------------------------------------

Introduction:

L. 29-40. This paragraph must be shortened by the authors.

Materials and Methods:

L.134. The broiler population (n=60) is very low in order to exhibit important and accurate results. The authors must exclude the in vivo experiment from this study.

L. 146. The lighting program usually is 24 h light during Day 1, then 23 h light and 1 h dark until day7, and from day 8 to 10 is set to 18 h light and 6 h dark.

Statistical analysis:

L. 159-166. Did the authors check the normality of the examined data by using nonparametric tests?

Results:

----------------------------------------------------------

Discussion:

L. 280-284. Authors must exclude the in vivo experiment due to the small (n=60) number of broilers.

Conclusions:

----------------------------------------------------------

Author Response

Dear authors, this study provides new and important data about the chemical characteristics and nutraceutical properties of Moringa peregrina (MP). Moreover, the use of the English language in the present study was appropriate and only some phrases must be rewritten. However, authors must answer several questions according mainly to the zootechnical and secondly to the statistical methods used. Firstly, regarding the in vivo experiment which was executed in the present study only n=60 birds were used, and this number of broilers is not appropriate to exert accurate results. Secondly, in the statistics section authors must answer if they check their data for normality with no parametric tests. Conclusively, the in vivo experiment must be excluded, and this study needs to be major revised to be published in this journal. The questions which must be answered and the changes which are proposed to be done are presented line by line in the following paragraphs.

Au: thank you for your very nice and precise comments and suggestions. Regarding the number of animals, also considering the available literature, is not too low. We have two groups only and for each group six replicates of 5 male animals. Looking at sample size determination   https://www.stat.ubc.ca/~rollin/stats/ssize/n2.html /,  using mean final body weight of the experimental groups, desired power of 0.95 and 0.05 probability level and other research, the number of the replicates as the experimental unit is adequate. The number or replicates  and sex of animals (males) for homogeneity  as well and were reported in the text.

Regarding the statistical analysis you are right, we performed the tests and added an indication in the material and method section.

The in vivo experiment is part of this trial because it aims not to evaluate the animal performance of health, but the protein values of the diets including or not MP and the MS displayed the chemical composition, antioxidant properties, ME, protein quality and apparent and true amino acids utilization.

Introduction:

  1. 29-40. This paragraph must be shortened by the authors.

Au: The period has been shortened by 3 lines

Materials and Methods:

L.134. The broiler population (n=60) is very low in order to exhibit important and accurate results. The authors must exclude the in vivo experiment from this study.

Au: Thanks a lot, please see previous answer

  1. 146. The lighting program usually is 24 h light during Day 1, then 23 h light and 1 h dark until day7, and from day 8 to 10 is set to 18 h light and 6 h dark.

Au: you are right, this is the program in the European Union. There are not similar indications in Saudi Arabia, we will consider this in future experiment.

Statistical analysis:

  1. 159-166. Did the authors check the normality of the examined data by using nonparametric tests?

Au: done and added

Discussion:

  1. 280-284. Authors must exclude the in vivo experiment due to the small (n=60) number of broilers.

Au: Thank you very much, please see the previous answer.